# Dense and distributed neuropeptide network in the nerve net of *Hydra vulgaris*

**Johanna De La Cruz Rothenfusser**[ORCID][1]*, **Luis Alfonso Yáñez-Guerra**[ORCID][2], **Felix Teufel**[3], **Rafael Yuste**[1]

1 Neurotechnology Center, Department of Biological Sciences, Columbia University, New York, New York, United States of America, 2 School of Biological Sciences, University of Southampton, Southampton, United Kingdom, 3 Department of Biology, University of Copenhagen, Copenhagen, Denmark

* delacruzjohanna11@gmail.com

## Abstract

Neuroscience has long emphasized synaptic transmission and physical wiring as the substrate of brain function and behavior. However, an additional layer of connectivity — a "chemical connectome" formed by neuropeptide-GPCR signaling — has been increasingly recognized in animals such as *C. elegans*, *Drosophila*, and the cnidarian *Nematostella vectensis.* To further explore neuropeptide networks in basal metazoans, we analyzed the genome and transcriptome of the freshwater cnidarian *Hydra vulgaris*. *Hydra* offers unique experimental advantages: a simple nerve net, robust regenerative capacity, a well described behavioral repertoire, and tractable whole-body calcium imaging that allows mapping of neural and muscle activity, and cell type identity, in an integrated manner. This makes *Hydra* a powerful system to investigate how neuropeptidergic signaling shapes neuronal ensembles and behavior. Here, we identify 61 putative unique neuropeptides and 65 neuropeptide-specific G protein-coupled receptors (GPCRs). We show that different neuronal cell types display specific neuropeptide and receptor expression profiles, suggestive of defined communication pathways within *Hydra´s* decentralized nervous system. Network topology analysis of the neuropeptide network reveals a dense and distributed signaling architecture, with ectodermal neurons acting as centralized hubs for organism-wide coordination. Computational simulations using a simplified model of the nerve net demonstrate that this architecture can implement stable dynamical states. Our study reveals a comprehensive neuropeptidergic network in a non-bilaterian species, highlighting the evolutionary continuity and functional relevance of wireless chemical networks for complex behavior. Moreover, the distributed and recurrent connectivity we uncover suggests the existence in nervous systems of attractor neural networks implemented with chemical signaling, as opposed to synaptic wiring.

**Data availability statement:** All data are either uploaded or provided in the linked GitHub https://github.com/jd4068/Hydra_Connectome/tree/main. Original single-cell data used can be found at https://datadryad.org/dataset/doi:10.5061/dryad.v5r6077.

**Funding:** This work was supported by the National Science Foundation (grant 2203119 to R.Y) and the Vannevar Bush Faculty Award from the Office of Naval Research (grant N000142012828 to R.Y.). The funders had no role in study design, data collection and analysis, decision to publish, or preparation of the manuscript. Johanna de la Cruz received a salary from the second funding source.

**Competing interests:** The authors have declared that no competing interests exist.

## Author summary

For more than a century, neuroscience has focused on connections between neurons via synapses as the main basis of brain function. In this study, we explore an additional and less understood layer of neuronal communication: chemical signaling, mediated by small diffusible molecules called neuropeptides. Neuropeptides are conserved across organisms and also present in humans and do not require synaptic wiring and instead can act more like radio broadcast messages, reaching many cells at once, and activating only those with the right receptor. We chose the simple freshwater animal *Hydra* to study this form of communication because its nervous system is small, accessible, and can be studied with whole-body imaging and genetic analysis. By analyzing its genome and gene expression, we identified a rich set of neuropeptides and their matching receptors, revealing an extensive chemical communication network. We found that different neuron types use distinct combinations of these signals, suggesting organized communication pathways despite the absence of a centralized brain. By modeling this network, we show that such chemical signaling alone can support stable patterns of activity, similar to the neural networks thought to underlie behavior and memory in more complex brains. Our findings suggest that chemical communication is an ancient and fundamental principle of nervous system organization, providing a complementary framework to synaptic wiring for understanding how brains generate behavior.

## Introduction

There are two primary models of how nervous systems mediate communication. The traditional model, developed through the foundational work of Cajal [1] and Sherrington [2], posits that information is transmitted via chemical synapses between neurons, forming fast and directed circuits. In contrast, an alternative model proposed by Weiss [3] suggests that neurons can also communicate through non-synaptic mechanisms, including paracrine and endocrine signaling [3]. In this model, secreted chemical messengers diffuse through tissue or body fluids and activate target cells that express matching receptors—without requiring direct synaptic contact [4].

This form of "wireless" communication, described in *C. elegans* as well as *Drosophila* [5,6], is thought to be predominantly mediated by neuropeptides—short polypeptide hormones produced through the proteolytic cleavage of larger precursor proteins [7]. These peptides are often post-translationally modified, most commonly through C-terminal amidation [8], which increases their receptor-binding affinity and half-life. Neuropeptides can bind to G-protein-coupled receptors (GPCRs), triggering downstream signaling cascades that may alter cell states and

modulate behavior [9]. Across metazoans, this neuropeptidergic signaling system plays critical roles in neural plasticity, homeostasis, and developmental transitions [10,11].

Neuropeptides have been identified in all major animal lineages—including Bilateria, Cnidaria, Ctenophora, Placozoa, and even Porifera [10]. Remarkably, they are found in some species that entirely lack neurons, suggesting that neuropeptidergic signaling predates the origin of the nervous system itself [12,13]. As the sister group to Bilateria, Cnidaria provides a unique phylogenetic insight into the evolution of nervous systems. Cnidarians diverged more than 600 million years ago and possess a diffuse nerve net without centralized brains or ganglia. Yet they exhibit a surprisingly sophisticated behavior and full complement of neural cell types—including sensory, ganglion, and neurosecretory neurons—and a rich repertoire of both conserved and lineage-specific neuropeptides. These features make cnidarians ideal for investigating the molecular and cellular mechanisms of neural circuits and the roles that paracrine and neuropeptide driven connections play within them.

Within Cnidaria, the hydrozoan *Hydra vulgaris* stands out for its simplicity and experimental accessibility. Unlike other cnidarians, *Hydra* has no medusa stage and maintains a polyp form throughout its life. Its body consists of two epithelial layers—ectoderm and endoderm—separated by a mesoglea and surrounding a central gastric cavity [14]. The nervous system contains approximately 200–1,000 neurons, depending on the size of the animal, organized into two distinct nerve nets in the ectoderm and endoderm, correspondingly [15]. Despite this anatomical simplicity, *Hydra vulgaris* displays a wide range of behaviors, including prey capture, feeding, and complex locomotor actions like somersaulting and inchworming [16]. Previous studies have shown that individual neuropeptides can be sufficient to drive somersaulting, underscoring the functional importance of peptidergic signaling in this animal [17]. Past work has indeed demonstrated that *Hydra* expresses a broad array of neuropeptides—including members of classical neuropeptide families such as the RFamide and GLWamide [18,19]—as well as a large complement of putative GPCRs coding genes [20]. These molecules exhibit specific expression patterns across neuronal subtypes, suggesting the presence of a modular and potentially combinatorial code of neuropeptide communication. However, the full scope and structure of this network remain unknown.

Recent work analyzing *Hydra´s* transcriptomic data obtained from different body segments has revealed the cellular diversity of its nervous system, characterizing eleven transcriptionally distinct neuronal cell types that are continuously renewed from interstitial stem cells [21,22]. This ability to continuously regenerate its entire body, including its nervous system, from just a small fragment of tissue, is another fascinating aspect of *Hydra* as a model organism [23]. Neurons are not only maintained throughout the animal's life via ongoing differentiation from stem cells but are also regenerated *de novo* following injury or amputation. During regeneration, new neurons are specified in appropriate numbers and spatial distributions, and they integrate into existing neural circuits to restore behavior [24]. This makes *Hydra* a unique model to study how regenerative nervous systems form, reorganize, and regain function. It also allows for the investigation of how stable ensembles of neurons—and the neuropeptidergic communication patterns between them—are re-established over time. By tracking gene expression dynamics and activity patterns across regeneration, one can begin to ask how long-range coordination, signal integration, and functional connections emerge in the absence of fixed wiring. *Hydra* thus allows researchers to explore plasticity and self-organization in diffuse, non-synaptic neural architectures.

In this study, we leverage computational analyses of genomic and transcriptomic data to identify 61 putative unique neuropeptides and 65 candidate GPCRs in *Hydra vulgaris*. We characterize their expression across defined neuronal subtypes and infer a dense web of potential ligand-receptor interactions. This inferred signaling network exhibits topological organization, with particular enrichment in ectodermal neurons, and reveals possible hubs for communication between epithelial layers. We also model this network and show that it can sustained dynamical stable states. Together, our results support the view that *Hydra* implements a chemically rich "wireless connectome"—a neuropeptidergic signaling architecture capable of coordinating complex behaviors without synaptic wiring.

## Results

### Identification of neuropeptides in *Hydra´s* transcriptome

To understand the network of peptidergic signaling in *Hydra vulgaris*, we first sought to identify the complete set of neuro-peptides expressed in the organism. To do so, we carried out a comprehensive bioinformatics analysis of *Hydra´s* transcriptome and identified 16 distinct neuropeptide precursors encoding a total of 61 neuropeptides.

Identification of the neuropeptides began with the detection of signal peptide sequences, followed by the extraction of corresponding transcripts from the transcriptome (see Methods). Within these transcripts, dibasic cleavage sites were located to predict the final cleaved peptides. Precursors displayed considerable variation in their peptide content, with some encoding only a single neuropeptide, while others encoded up to ten distinct neuropeptides, with an average of three peptides per precursor. These precursors arose from 16 different genes within *Hydra´s* genome (S1 Table) and encode 5 well-characterized peptide families, as well as some novel or less studied peptide types. Most predicted peptide precursors contain a clear signal-peptide sequence and more than two cleavage sites, showing that often multiple cleavage sites can be found surrounding a final peptide. Overall, our search identified 61 candidate peptides (Methods). These were subsequently manually examined and cross-referenced with known cnidarian neuropeptides from the literature. The three genes that coded for more than one transcript were from the characterized peptide families RFamide, PRXamide and KVamide [25,26]. These were also the families with the highest number of mature peptides. Furthermore, the GLWa-mide gene had the highest number of peptides derived from a single transcript, with 10 predicted peptides coming from the same precursor (Figs 1 and S5). Furthermore, we also identified less-studied peptide precursors such as the ones encoding the peptides, GGYGYamide, phoenixin and MIE amide. While some of them have been described in previous studies their roles in neuromodulation are still unclear [10,28,29]. These peptides' patterns occurred at lower frequency, with some transcripts encoding only one final peptide (Fig 1).

### Evolutionary conservation of neuropeptides within and across clades

As an evolutionary mechanism, the use of neuropeptide precursors to generate peptide actuators provides two key advantages: it allows a single protein precursor sequence to encode a diverse array of neuropeptides, generating different products depending on cleavage patterns, or, alternatively, it enables the production of multiple similar peptides, ensuring a rapid and high-yield response to stimuli that trigger neuropeptide release. Both mechanisms are conserved across different types of peptides, including cyclic peptides and hormones. Conserved patterns across peptide families found in *Hydra* also have been found in other organisms across clades. For example, RF, PRX and GLWamides are shared across all cnidarian classes except for Polypodiozoa and Myxozoa, where no PRXamide precursors have been detected in some species [18]. Other peptides that we found in *Hydra*, such as RFamide and LRWamide, are only shared among certain groups within cnidarians, Hexacorallia and Anthozoa respectively [18]. Although most neuropeptides are thought to have evolved after the separation of bilaterians [4] there is evidence of some hormone-related peptides as orthologues between both cnidarians and bilaterians. For example, phoenixin, a peptide found across several metazoan clades [10] – is a peptide detected in mammals and mice where it is thought to play a role in appetite control and mating in mammals and recently also in a goldfish model [30]. These findings further support a common neuropeptidergic ancestor system of cnidarian and bilaterian [10,31]. Moreover, the widespread conservation of neuropeptides found in the literature highlights the evolutionary stability and functional importance of these signaling molecules [10,32]. The diverse lengths and sequence patterns of the neuropeptides identified in *Hydra vulgaris* also pointed to specialized roles in cellular signaling and coordination across different cell types.

### Identification of G-protein coupled receptors in *Hydra´s* genome

We then proceeded to identify putative receptors for *Hydra´s* neuropeptides. As in other species [11], we assumed that G-protein coupled receptors (GPCRs) mediate most neuropeptide signaling, so we focused on this family of proteins. We

>>t1679aep.1_split.1/HVAEP1.T016173.1 (Hym172/176/357690) Hym-176 Family
MSKINKLTMYVFYAFLVLNIYVVLSVNSLPLRDDEDTDEIDGDISELENEYQTNQVYDYNKFKNQADLKVKSRNHYAPFIFPGPKVGRDVNFHSVLSPSDESRKSFNT
YYENGYQHDKPAFLFKGYKPGDQTQKNL
> t2059aep.2_split.1/HVAEP9.T017227.1 (RFamide, RFamideII, RFamideV, RFamideVII, RFamideVIII) RFamide Family
DPVEKLKMLSNKKVKLLFALVLIVVEVVKSDDKNFSLEVNKDVKRFIKDILDAKSEEQLMSGRFGKSLPDEEDIDNEVENEYDNEYDDETESQGIINGRYGRQLLRGR
FGRQNDNKAASKESQWLGGRFGKEVATQWFNGRFGREIGGRFLPRFGREFNKPHYRGRFGRVAKL
> t3809aep.2_split.1/HVAEP1.T012292.1 (RFamide I , RFamideII, RFamideIII, RFamideV, RFamideVI) RFamide Family
KKKVKMLSNKKVELLFALVLVVVAVVRSEDKNLLLEDNKDVKRIVNDYLETKNGEQLMSGRFGKRETDEGDSDDEDSSEYENEYDDELENQGLANVRYERQLMRG
RFGREKNAVSNEDQWLGGRFGREAATQWFNGRFGRDIEGRFLPRFAKESNKPHLRGRFGRAAKL
>t7664aep.3_split.1/HVAEP1.T004115.1 (Hym-355) Hym-355 Family
MMRTAVFGCFILFTIVLALPYRDAFDLFDRFDEYIEKVAKVTADEARLLRDVRNFYKLTKENFVSNADEDDFQDYAPRGGKRENRPRPGK
>t11055aep.1_split.1/HVAEP1.T018128.1 (Hym-53/54/248/249/331/338/370/1071) GLWamide Family
MGMFERKKIVLLVSLICVSQQAANVQDANSKSTSTELKVVKPQKRVTPVKDAEKLSILRTQDNSLDLNTNGEEVVWDELTHNIPLEYIEKIYNELNQLAQNENRPKRL
WGATAAINTDNLNPEVENELENKKNAPVIEKFERPIGLWHKDVETKNPENRLPLGLWGKDSEPLPIGLWGKDADVNDDLKKEPLPIGLWGKDIDSTQEDNKPNAY
KGKLPIGLWGKDNALTNDFGKKNNGKDSGPPPGLWGKDSKPIPGLWGKDNGPMTGLWGKKDVGPPPGLWGKKDQPPIGMWGRAGKKDSNPYPGLWGKKE
EEIENVDKEFKEDSLEEYPACLFENPPCEIQEKRYKIEKSGPPPGLWGKRSEKYSMNKPPWRGGMWGRSEILENSVHDSKQTNTIDMEHAEN
>t12588aep.1_split.1/HVAEP1.T016171.1 Hym-176 Family
QVFIVQLKKKKMSKANKLTAFNILLVLNIFVILAVNSLPLRDDEEIDSEIDGDITELENGYQNTQINSYDRHKKQLNPKDKNKKFMIFQGPKVGRDVDFHSVQSPSNK
VGKSTRFYYGNDYR
>t12874aep.2_split.1/ HVAEP2.T004115.1 (Hym-355) Hym-355 Family
KMLSLTVATLLLITSIVMAMPNRDATDSNESDILNILDEYIVKVAEMTANEAKILNDVRNYYNDRSSKSLGEFPQSFLPRGGKRDARPRAGK
>t16657aep.3_split.1/HVAEP1.T017226.1 (RFamideII, RFamideIII, RFamideV) RFamide Family
KVSECNHHQVKKKVKMLNHKIETLLVWGLIIVAVVKSEDKNLSAEDRKDVKRIVKDYLNIKNGEQLMSGRFGKRVTDEDIDNEIESEYENEYEDELENFANGREDAA
QWFNGRFGREIGGRILPRFATESNKPHLRGRFGRAAKM
>t17992aep.2_split.1/HVAEP1.T016170.1 (Hym-357/690) Hym-357 Family
NEGKMSKVKKLCEFNIILVLYIFLVFSVNALPFKDDEETGIEFDGNISESGNEYQSNQYYDYNKIKNQIYNDYPNIIEKNFKPLKVMKMGRGANDHFDQIGSRKSNDV
NLINGNQQDKPAFLFKGYKPGDQTQKKS
>t21435aep.2_split.3/HVAEP1.T008452.1 PNG Family
ERKMTRATLAIFFLAILLVIIENNVADRKSHTRHPSIRPSKSVPNGGRPTSLKPSKSETSGRRTPSIKPSKAGGNGKRHSSIKPSKPVPIQSSRNIDSNRDKNKNHKDKK
SFGRGHKSVPNG
>t25706aep.1_split.1/HVAEP1.T017220.1  (RFamide I ) RFamide Family
KMATNMALLAFVFFATSIFMLTKADQNEDNQKYDGIARSLKVLLQNYNEKQEEKSDIQNIIEKFSEYQNTGKTIQRKDNVNPMFEKKDAVEQWLGGRFGRVVYDL
LLSEVSKDHKRNDETNPMIEKKDADTENRFNREALEQWFSGRFGLTNHKRNDEVNPMIEKKDSEIENRFNREAIEQWLGGRFGRTVYEFLLSETPEKRKK
>t25807aep.3_split.1/HVAEP1.T02115.1 (Framide1 also Hym-65, Framide2 also Hym-1533) RFamide Family
YEFKGFRPLTWKVDFYKVLCFILRNFLVHLETKMYLRLLLVFFVLQISLQESNVRQLDLGQLIEDYLAKENVRREEFLNKINTEILRYIYELENENKGKKRIEASADKNVL
EKVLTEVPSIRESVTSKESNVNKMHNSLDSKSSIRSIPTGTLIFRGKKESNSNNENASEQGAPGSLLFRGKKEPNVKENSKNETEASHGERLQQTERNFLVKTKEYIEK
LLNSGEEIV
>t33899aep.1_split.1/HVAEP1.T017227.1 (RFamideII, RFamideIII) RFamide Family
VKMLNQRKVEIFFALALIVVALVKSDDNNHLSEGSKNNHLSEDSKNIKRILKDYLNAKNAEQLTRGKLMKRITNKEDNFENDVENEYENKYDDELKNNGHVSKRED
ATQWFNGRFGREMGERFLPRFGKELNKPHLRGRFGRNIKL
>t6969aep.1_split.1/ HVAEP10.G018620 GGYGYamide Family
RWFMGRTKLFNVNLLAEAAPFKHAKDIIADTKNRDNDDEDESNREENKDLFSNKLSDYGGGYGYGKKEINEDEDQSEGGYGGYGGYGYGKRETERDLGYGGGYG
YGKKEINEVADPWRGGR
>t38444aep.1_split.9/HVAEP1.T026265.1 PW Family
KKNSQNMVFSIRLLLLVLILHFQIYKAEEQVSDNLKSDQTTNEIEELFLKNDVSDREKEKTLNKALNDLKTILNDNYDNYNKRNDKINENQKIQKNMLSEEIQKKNEN
DNLKNGPNAALPWRKDELSMINSLLKRMESSGLLKNNVNNELKDNKKEELNSPALPWRREKLTINSLFNRMESLGLLKRNVNNDLKDPNAALPWRRDEVSLESLL
KLMESIGLHKNNKNNDLKGPNAALPWRRDELSNESFLKRLKSSGLFKDNENKELKSPALPWRRDGLSIESLLKRLESFKDNKNNEFNSPALPWRRDEISIKSLLNKM
ESLGLLENNENDELKSPALPWRRDEISIKSLLNKMESLGLLENNENDELKSPALPWRKEELSIGSLLKRIELLGLLNENKNLHRGSLMLKKDENSFVNFPTKDSAILRDS
AIFDNLKDNTMLENEKEKKDNKNSNINDKIAELFKKFKDYFSTENSSETDF
>t21227aep/HVAEP12.T022823 Phoenixin Family
MSLISKLAVYGLVLATGLSLVPIYFVPKAVPEKYRDIQKVSRKDIVQSEVQPGNMKIWSDPFDRKK

**Fig 1. Identification and characterization of neuropeptide precursors and final products in Hydra vulgaris.** Amino acid sequences of 16 precursors identified encoding 64 final peptide-candidates via dibasic cleaving sites. Red shows an identified signal sequence detected by signalP(25), green a predicted final peptide, yellow identified cleavage sites. Underlined sequences are peptides verified experimentally using MS [27]. Identified versions are sometimes shorter or longer than found ones. Full list of transcriptome and genome position in S1 Table. Neuropeptide family assignments are indicated in cyan.

identified approximately 100 GPCR candidates in the *Hydra* genome and, using a cluster-based analysis (CLANS), we found a subset of these receptors forming part of a cluster that contains known neuropeptide receptors from other species [20], including the recently deorphanized receptors from *Nematostella vectensis*, the cnidarian with the largest number of deorphanized receptors ([20],Fig 2; Methods). For further analysis we selected GPCRs that statistically clustered close to experimentally deorphanized neuropeptide binding GPCRs in *Nematostella vectensis*, by using their published sequences [20,28]. This reduced our candidates to 65 GPCRs (highlighted in green Fig 2, see App. 2). The obtained predictions cluster into subgroups with a larger group having a high shared similarity with NPR and FLRN binding GPCRs from *Nematostella*. This is expected as the GPCR neuropeptide network seems to be conserved across species and is in line with the finding that several of the same neuropeptides seem to be used across different systems [32]. However, we also observe several clusters forming that seem to be *Hydra* specific indicating the possible expansion of GPCRs with novel binding affinities, echoing findings in nematode species of rapid receptor evolution across species allowing novel communication pathways [25]. We determine several non-neuropeptide binding GPCR clusters that represent other classic target groups such as monoamine and trace amine binding GPCR. Several predicted *Hydra* proteins also aggregate towards these

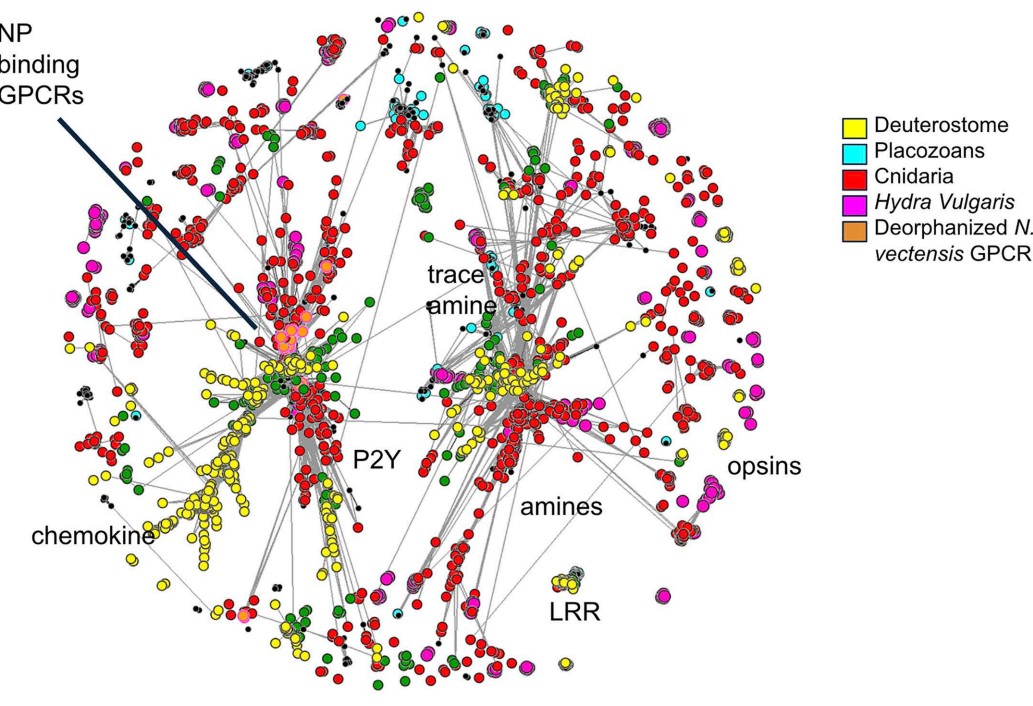

p value = 6e-38

**Fig 2. Identification and interspecies comparison of Hydra´s GPCRs.** Cluster analysis of major Class A GPCRs from cnidarian, bilaterian, and placozoan species, illustrating sequence similarity relationships. Each node represents a GPCR sequence, with edges indicating similarity-based connections. Different species groups GPCRs are color-coded. Orange highlights GPCRs sequences that have been deorphanized and shown experimentally to bind neuropeptides in Nematostella vectensis. These were used as a guidance to determine GPCR from Hydra vulgaris that putatively can bind neuropeptides (highlighted in green). Edges are colored by intensity. Fruchterman Reingold algorithm used for clustering with repulsion set to 40 and attraction between nodes set to 10.

PLOS Computational Biology

allowing further annotation of these receptors as well as the observation that we again have clusters with unique species identity. These sequences are available and could be of further interest to understand GPCR evolution and rapid adaptation to speciation, however this was excluded from this study as this was outside the scope of our analysis.

### Network of predicted binding interactions between identified neuropeptides and GPCRs

To reconstruct the neuropeptide chemical network *Hydra*, we then explored the potential interactions between identified neuropeptides and GPCR candidates. To computationally predict them, we systematically assessed chemical binding probabilities for every receptor-peptide pair. This analysis integrated prediction scores derived from AlphaFold [11,33,34] and Deep TMHMM ([35], see Methods), focusing on expected binding potential within membrane-localized regions [36]. The predictions revealed a broad landscape of potential interactions, with receptors displaying high scores for multiple peptides and vice versa (Fig 3). To reduce the rate of biologically inaccurate predictions and ensure meaningful predictions, only top high-scoring receptor-peptide pairs were selected for further analysis, thresholding the binding potentials,

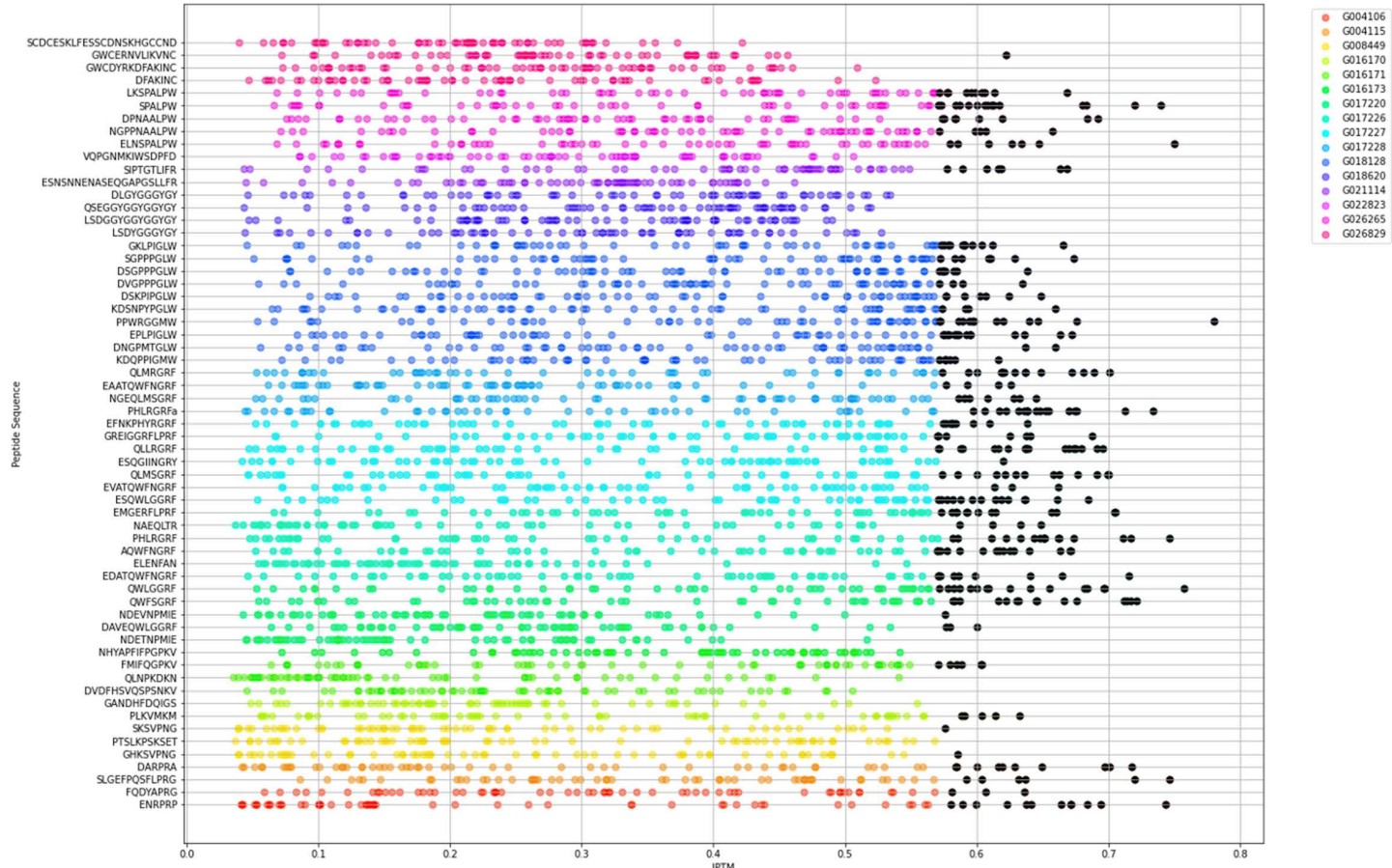

**Fig 3. Predicted interactions between identified peptides and GPCRs in Hydra.** Binding prediction results for identified peptides and GPCRs, with top 10% highest scores highlighted by black circles. Colors represent precursors from which the respective peptides are synthesized. Prediction scores integrate results from AlphaFold and DeepTMHMM to assess expected binding potential of the receptor-neuropeptide combination within the predicted membrane location (see Methods). Note that several different receptors can bind each neuropeptide. Further, several of receptors are found to bind more than one neuropeptide and have high scores even across families, reinforcing the idea that the neuropeptide network is flexible and relies on highly distributed set of interactions.

for the top 10%, based on the thresholds established by previous experimental verification [36]. This was based on previous studies that have shown that, although highest match predictions are accurate, they are not always the ones occurring *in vivo* [36]. Further, several studies have determined that a high level of promiscuity in GPCR neuropeptide binding [37] makes it necessary to postulate a model that simulates more than just the strongest and statistically most probable connections.

We then used these filtered pairs to build a *Hydra* neuropeptide-GPCR network, enabling the visualization and analysis of neuropeptide-mediated communication pathways. Out of 2,745 possible combinations, 433 receptor-peptide pairs passed the threshold. The filtering resulted in several predicted peptides to remain orphan, as they did not bind any receptor with sufficient confidence. This includes the GGYGYamide family as well as the phoenixin homolog, raising questions of their function in the cnidarian clade which could rely on very high expression amount that saturate and induce binding. Another possibility is that they might be binding different GPCR targets such as ion channels, as has been demonstrated in several studies [37,38] However, these assumptions must be validated *in vivo*. Interestingly, 49% of the selected pairs were formed by the RFamide peptide family, with many of the peptides in the family having strong binding scores with several receptors in the family. Following RF peptides, the second group with highest connections was the PWamide family, where one specific peptide (SPALPW) had 28 high binding predictions across the determined NP binding GPCRs (Fig 3). We also observed that some receptors were highly unspecific in their interactions, binding peptides across different families. An example is GPCR G010112 which interacted with 37 different peptides from families including GLWamide, FR and RFamide. This reflected a distributed nature of the topology of GPCR-peptide interactions commonly observed in neuropeptide signaling systems [39,40].

**Neuronal cell types express specific neuropeptide precursors but many putative receptors**

To explore the network structure of the predicted neuropeptide-binding GPCRs across neuronal subtypes, we analyzed the data on a cell-type basis, using *Hydra´s* single cell transcriptome database, which describes 11 neuronal cell types [14]. In this analysis, we found that each neuronal subtype exhibited a unique expression pattern of neuropeptides and receptors (Fig 4). This replicates findings from *C. elegans* [39] and *Drosophila* [40] where these specific topologies allow a higher degree of specialization and highly unique and context dependent communication pathways. Overall, the neuropeptide precursor expression across neuronal cell types *in Hydra* was quite specific, showing localized clusters of neurons in the ectoderm or endoderm (named by the letters "ec" and "en," respectively, followed by a number) showing similar expression patterns. Interestingly, neuropeptide precursor expression similarities were not selective to endoderm – ectoderm identities, as the ectoderm neuronal subtypes ec1A, ec1B and the endoderm subtypes en2 and en3 had similarly high expression levels of the precursor of the FRamide peptide, for example. We also observed similar precursor expression of GLWamide and PRXamide peptide family across the ec3 neuronal subclusters, showing that, although they might have differential expression patterns when considering the final peptide, they likely shared a developmental identity. Within this overall specific yet distributed expression pattern of neuropeptides precursors across neuronal cell types, some neuronal types exhibited particularly high specific expression of precursors, with, for example, only ec2 expressing the precursor of RFamide and PMIEamide peptides and en1 having very high expression of the PRX amide precursors.

Complementing the specific and distributed expression of neuropeptides across neuronal cell types, when examining receptor expression patterns, we observed a much broader spread of expression, with most neurons expressing GPCRs at moderate to low levels of expression, when compared to the high and unique expression clusters observed in the neuropeptide matrix. An exception of this was the group of ec3 neurons, which showed a very high expression of two GPCRs that were not found in any other cell type. Furthermore, subsets of these neurons expressed this GPCR in a distinct manner potentially supporting a specific communication pathway between them.

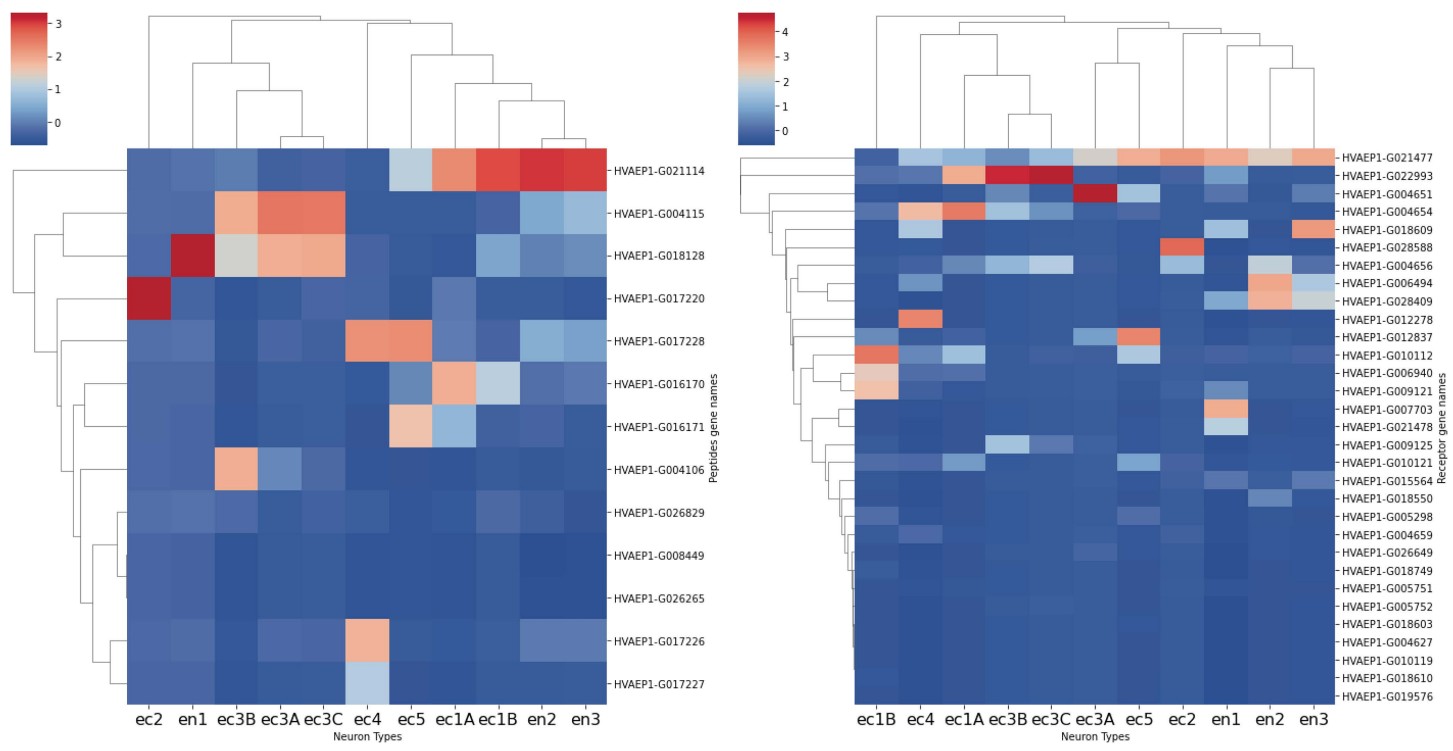

**Fig 4. Specific expression of predicted neuropeptide and GPCR genes in Hydra´s neuronal subtypes.** Transcriptomic profiles of predicted neuropeptide and neuropeptide-binding GPCR genes across neuronal subtypes in Hydra. X axis categorizes different transcriptomic neuron types found in Hydra and on Y axis detected Neuropeptide (left) or GPCR (right) transcripts. Note that there are only 16 neuropeptide transcripts - these encode the final 65 detected peptides. The color shows the average expression across cells in those categories. The unique expression profiles for different neuronal subtypes suggest highly specific communication pathways within the organism. The cladograms show similarity between cell types based on receptor and or prepropeptide expression.

## Dense and recurrent neuropeptide-GPCR network

To better understand the network architecture of neuronal communication, key neuronal subtypes were visualized in a network representation, where nodes represent neuronal subtypes, and edges correspond to communication pathways (Fig 5). We first visualized the connectivity of this network in a matrix format (Fig 5A), where each neuron is represented as both a row and a column. The connectivity values were derived from receptor-peptide interaction scores, incorporating both receptor and neuropeptide precursor expression levels in interacting cells (see Methods). A striking feature of this matrix was the highly interconnected nature of specific neuronal subtypes expressing complementary sets of neuropeptides and putative receptors. This is reflected in a density of 0.47 (methods) which is high for a biological network, yet consistent with past chemical connectivity analysis [5]. This high density also is further accentuated by around half the neurons forming positive feedback recurrent "self-loops," thus potentially activating their own pathways in response to their neuropeptide release. On average, neurons displayed a high connectivity (mean weighted degree ~20,000), highlighting the dense and highly interconnected nature of the network. Notably, degree distribution analyses (see S1 Fig) reveal distinct patterns between incoming and outgoing degrees. The incoming (weighted in-degree) distribution was broad and highly variable, indicating diverse levels of signal reception among neurons. Conversely, the outgoing (weighted out-degree) distribution was bimodal, suggesting that a subgroup of neurons forms a highly interconnected cluster. This subgroup may represent specialized hub-like neurons that orchestrate signaling or regulate local network dynamics that support complex signaling interactions, including potential self-regulatory feedback loops. There was also

PLOS Computational Biology

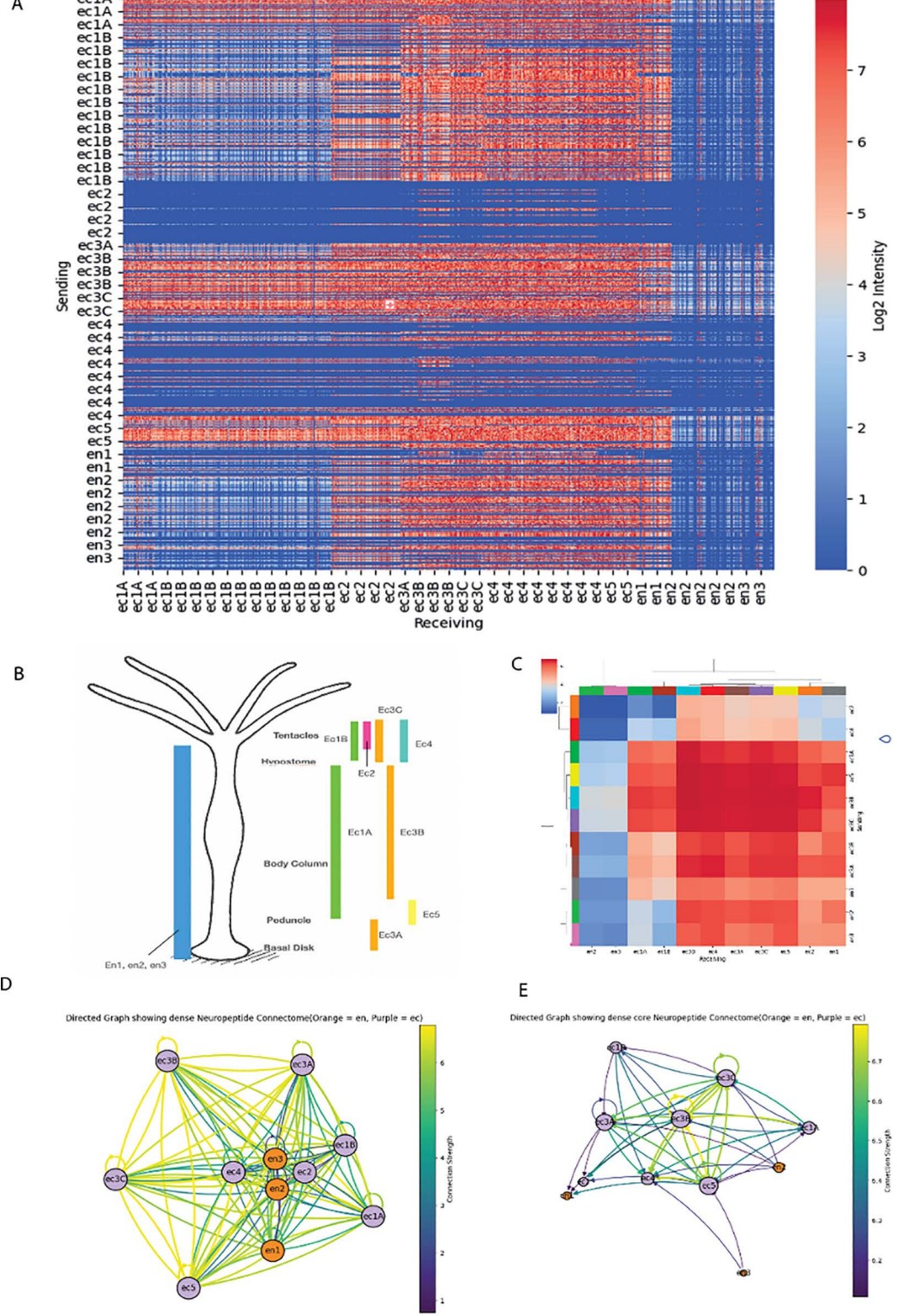

**Fig 5. Dense neuropeptide-GPCR network.** Network derived from predicted neuropeptide-GPCR interaction, showing hub nodes within and between cell types. (A) Cellular matrix, where rows denote sending neurons, while columns denote receiving neurons. Communication strength is color-coded, with red indicating high intensity and blue indicating low intensity. Strength is calculated based on number and expression levels of GPCR-neuropeptide

combinations. Note widespread high density of interactions. (B) Spatial distribution of 11 neuronal subtypes along the Hydra´s body (adapted from [21]. (C) Neuronal cell type-level network showing strongly connected ectodermal communication hub, with clustering based on connection similarity. (D) Topology of neuronal interactions via neuropeptides/GPCRs, illustrating a densely connected network with multiple communication pathways between cell types. Node placement is determined by directed force algorithms based on connections. Endodermal and ectodermal neurons are colored blue and pink, respectively. (E) Simplified version of D with strongest 30% of connections, highlighting ectodermal hub nodes.

a difference in the degree of connectivity between endodermal and ectodermal neurons with the later having a higher degree of connectivity (S3 Fig).

## Ectoderm nodes dominate highly interconnected network

We also further investigated the interconnectedness of our network. To do this we subsetted a random group of neurons (10% of the original) which would be expected in the animal. Contrary to classical network graphs, where a rich club is typically formed by a small subset of highly connected nodes, we found that the network was particularly well connected, with a rich club formed by over 100 neurons (based on the degree thresholds shown in the Rich Club Coefficient plot). For example, at a degree threshold of approximately 75, the Rich Club Coefficient remains high (close to 0.9), indicating that these neurons are highly interconnected among themselves. At higher thresholds (e.g., 250+), the coefficient gradually declines but remains substantial (0.6–0.7), suggesting a distributed network of hub-like neurons even among the highest-degree nodes (S3 Fig). The composition of the Rich Club at the top 10% threshold revealed a clear bias towards ectodermal neurons (ec), which make up the majority (96 of 101 nodes), with only a few endodermal neurons (en, S4 Fig). This composition highlights the role of ectodermal neurons as key communication hubs in Hydra´s neuropeptide network.

This analysis also revealed a clear heterogeneity in the connectivity of different cell types, suggesting that there might be different roles within neurons of different populations. This was particularly clear in ec2 and ec4, where only two of neurons seem to be sending signals while the larger part of the population was only receiving signals (Fig 5A). We summarized the results of this matrix at the level of broader neuronal clusters (Fig 5C) and further visualized it as a network graph (Fig 5D). This graph revealed important communication hubs, where different neuronal subtypes seem to be having specific roles. For example, neurons of the ec1A and ec1B subtypes exhibited dense connectivity, particularly with ec2, ec3, and ec4, suggesting a highly active signaling network. In contrast, neurons of ec4 and ec2 subtype primarily function as receiving nodes, with minimal outgoing signaling. When generating a graph with only the strongest connections, a network core emerged (Fig 5E), centered on ectodermal neurons, reinforcing their role as primary conduits of information.

## Extensive endo-ectodermal neuropeptide communication

We used the reconstructed network of neuropeptide-GPCR interactions to characterize the signaling pathways between the ectodermal and endodermal nerve nets of Hydra. These two nerve nets are separated by the mesoglea and show low to no demonstrated synaptic connectivity with each other, and ultrastructural reconstructions reveal a lack of neuronal processes crossing the mesoglea [4,17,41,42]. This presents a potential conundrum when trying to understand how the two nerve nets work in a coordinated fashion to generate muscle activity and behavior [16]. We addressed this issue by analyzing putative neuropeptide interactions between ectodermal and endodermal neurons, finding multiple reciprocal signaling pathways, that demonstrate the bidirectional nature of endoderm-to-ectoderm chemical communication. Specifically, we identified 20 signaling connections from the endoderm to the ectoderm and 15 from the ectoderm to the endoderm (Fig 6). Although ectodermal neuronal populations generally sent and received signals with similar strengths, endodermal neurons exhibited clear functional differences, with, for example, en2 functioning primarily as a receiving neuron that integrates signals from multiple ectodermal sources, while en1 acts as a primary sender relaying information to ectodermal neurons. When modeling these interactions as a

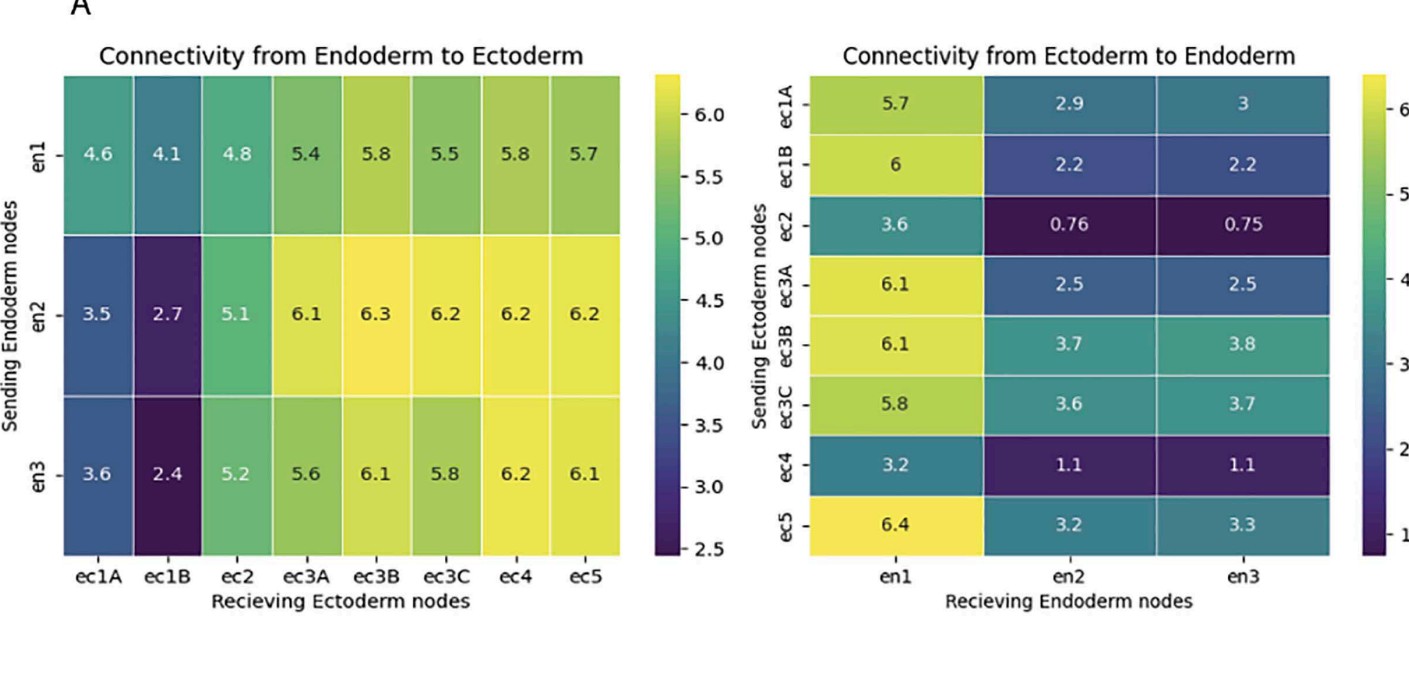

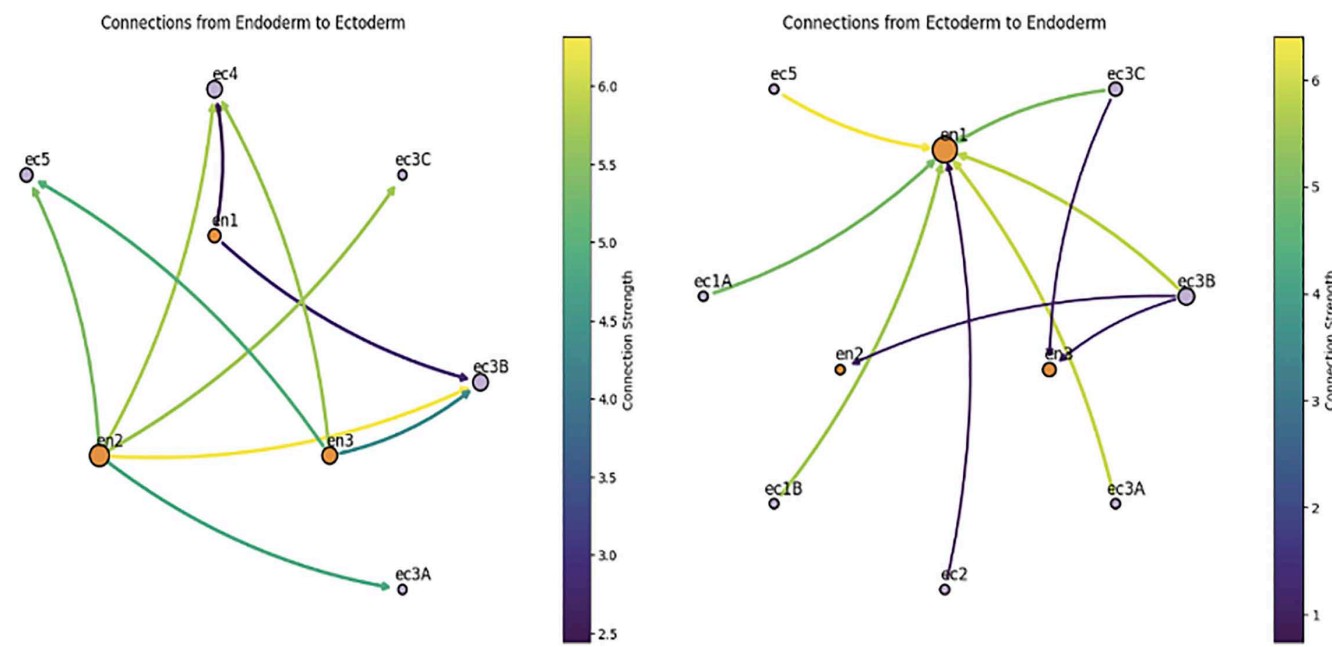

**Fig 6. Connectivity between endoderm and ectoderm nerve nets.** A. Heatmaps showing directed connectivity strengths between endoderm and ectoderm nodes. Left: Connectivity from endoderm to ectoderm nodes. Right: Connectivity from ectoderm to endoderm nodes. Color scales represent connection strength, with higher values indicating stronger connectivity. Each row represents a sending node group, and each column represents a receiving node group. Color bars indicate the range of connectivity strengths. B. Rich neuropeptide connectivity between endoderm and ectoderm. Left: Connections from Endoderm to Ectoderm nodes. Right: Connections from Ectoderm to Endoderm nodes. Edges are colored according to connection strength (see color bars), with node labels indicating node groups (e.g., en1, ec5). Node sizes are proportional to their overall connectivity strength. A variety of connections between endoderm and ectoderm are shown, with different cell types exhibiting specific subfunctions highlighted by the reduced figure. Note that en1 is a receiving hub for ectodermal signals, while en2 sends strong signals towards the ectoderm.

network, distinct hubs emerged, with ec3B, ec5, and ec3A forming the major ectodermal signaling nodes, exhibiting both high outgoing and incoming connectivity, while en1 and en2 act as key intermediates linking endodermal signals to ectodermal processing centers (Fig 6).

### *Hydra´*s neuropeptide network can implement stable dynamical states

To explore potential function of the dense and recurrent neuropeptide network and to assess if it was capable of leading to differential states of activity, we analyzed how the network would react to different inputs. To test this, we implemented a minimal recurrent dynamical model using the inferred interaction weights from our network analysis and a standard nonlinear activation function. Simulations of this model revealed that the network reliably converges to stable equilibrium patterns, as attractors, and that different initial conditions can also lead to different final states (Fig 7). We then tested if different inputs could be represented as different initial states that the network, exploring the dynamical behavior of this model network over time, by characterizing the dynamics of the en1 node across 3000 timesteps with different initial conditions. We observed two stable states for this node, which demonstrates how the node adapts to a changing environment (Fig 7A). A similar bi-stability was observed for the entire network when representing change and final state of the network as PCs and energy landscapes, with two distinct final states corresponding to the two different inputs (Fig 7B, 7C). The complexity of these states will likely increase if we simulate with the entire network, but these results already demonstrate a network capable of reacting to different environmental stimulation by settling in different dynamical trajectories.

## Discussion

### A dense and distributed neuropeptide network in *Hydra vulgaris*

In this study we perform a computational analysis of the genome and transcriptome of *Hydra vulgaris* to identify its potential complement of neuropeptides and its receptors. We find 16 precursors, 61 putative peptides and 65 putative GPCRs that bind them. The single cell transcriptome data reveals that neuronal cell types normally express selective sets of neuropeptides, and a broad set of putative receptors. These results are consistent with data from other organisms [13] that show specific neuropeptide-GPCR pair interactions triggering complex behaviors. Using computational and AI-based methods, we then calculate the binding scores of all possible neuropeptide-GPCR interactions to reveal a network of potential chemical interactions. This network is dense and distributed, meaning that different connectivity patterns are found and some neuropeptides can interact with several receptors and vice versa, while others seem to be more specific or less common targets. Network analysis also reveals the prevalence of feedback interactions made possible through the by the highly recurrent connectivity, to the point of several neuronal cell types stimulating themselves. The topology of the network shows the presence of neurons that act as hubs that centralize the flow of interactions and likely represent key components of *Hydra´*s neuropeptide signaling system, potentially mediating critical physiological and behavioral processes. To assess the functional implications of this topology, we implemented a minimal recurrent dynamical model using the inferred interaction weights. Simulations converged onto stable equilibrium patterns, and different initial conditions led to distinct final states, demonstrating that the network is dynamically capable of multi-stability. This indicates that *Hydra*'s neuropeptide network is not only structurally complex but also capable of supporting multiple stable signaling modes. Overall, these results show that *Hydra* neurons communicate through a structured and non-random neuropeptide signaling architecture, combining localized specificity with a small set of influential hubs. The patterns suggest that *Hydra* neurons communicate through highly specific pathways, where the complementary expression of certain neuropeptides and receptors mediates targeted, localized signaling. This organization provides a previously unrecognized level of hierarchical and dynamic structure in a basal nervous system.

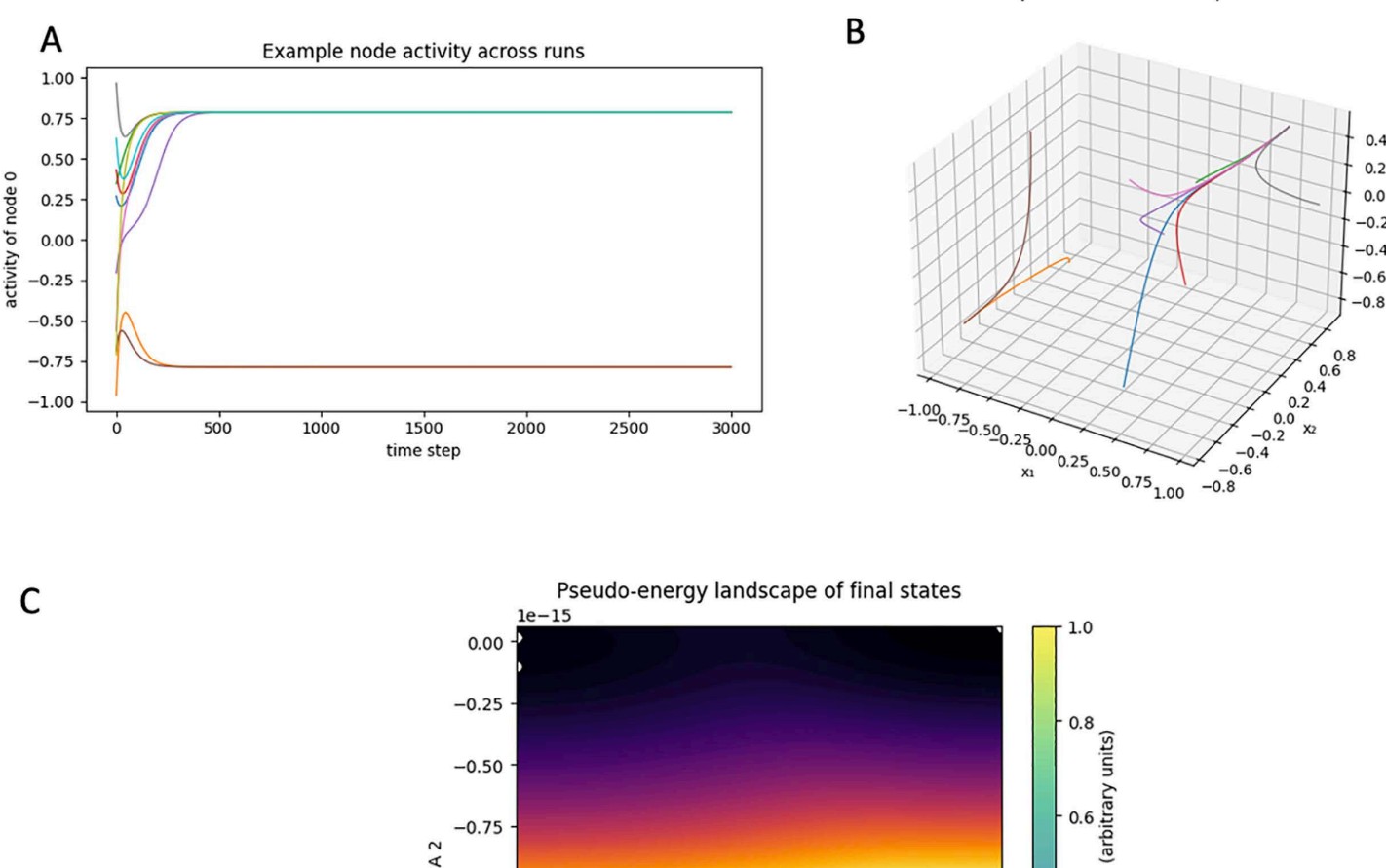

**Fig 7. Hydra´s neuropeptide network can implement attractor states.** A. Activity of one of the network nodes (en) across different initial starting points demonstrating a dynamical split into two different final states depending on the initial input towards the network. B. Representative trajectories of the recurrent dynamical system starting from different initial conditions, projected into the first three principal components of the network's state space. Each trajectory is shown in a different color. Despite diverse starting points, trajectories converge toward two distinct terminal regions, reflecting the multistable nature of the system. Axes correspond to the reduced state-space dimensions ($x_1$ = PC1, $x_2$ = PC2, $x_3$ = PC3). C. Shows a pseudo-energy measure for the final steady states reached from a range of initial conditions. The resulting landscape is visualized in the reduced state space defined by the first two principal components (PC1 and PC2). Brighter regions indicate lower relative pseudo-energy (more stable attractor basins). The landscape reveals two distinct basins of attraction, consistent with the presence of multiple stable equilibrium states supported by the dense and recurrent architecture of the neuropeptide signaling network. Axes correspond to the reduced state-space dimensions ($x_1$ = PC1, $x_2$ = PC2).

## Neuropeptide communication between *Hydra´s* two nerve nets

Our results reveal extensive neuropeptide-based chemical interactions between the ectoderm and endoderm neurons. This chemical network can help solve the riddle of how the ectoderm and the endoderm can communicate, when no neuronal process crosses the mesoglea that separates them [15]. Neuropeptides likely diffuse freely through the mesoglea, as their bath application can engage the nervous system of the entire animal [17]. In this neuropeptide network, ec1 neurons act as major signal senders, connecting to most other neurons but receiving highly selective input from ectodermal clusters. Interestingly, a broader pattern emerges when considering tissue-layer identity: endodermal neurons predominantly function as signal senders, whereas ectodermal neurons exhibit more diverse roles, with some acting as relays that both send and receive signals. This suggests that ectodermal neurons may serve as key intermediates in communication, integrating inputs from multiple sources and coordinating responses across the network. Moreover, the network suggests that endodermal neurons communicate with each other primarily through ectodermal intermediaries, rather than forming strong direct connections. This structured communication suggests that endodermal neurons may influence ectodermal activity more than previously thought, through non-synaptic signaling mechanisms such as neuropeptides. Ectodermal neurons appear to integrate these signals and relay them across the nerve net, ensuring coordinated movement responses, with en1's strong sending role suggesting it may function as a critical regulator of cross-tissue communication. Our data then suggest a model whereby chemical synapses may only be effective within each tissue whereas neuropeptides or mechanical coupling can serve to coordinate them and generate animal-wide behavior.

## Neuropeptide families mediate chemical communication in *Hydra´s* nerve net hubs

Our analysis indicates that communication between neuronal subtypes is not only highly specific but also structured around distinct neuropeptide families. Although we cannot yet pinpoint the exact neuropeptide responsible for each observed physiological response, the data reveal clear patterns of family-level specificity. One example includes the endodermal hub neurons en1 and en2, that show strong expression of a shared receptor (Figs 4 and 6). This suggests that these two hubs are coordinated and poised to respond robustly to incoming signals—potentially enabling rapid behavioral changes such as body extension or the initiation of feeding. A second example of specificity arises in ec3B, the major signaling neuron linking ectoderm and endoderm. Ec3B uniquely expresses a single precursor encoding PRXamide-family peptides (gene *G004106;* S5 Fig). This exclusivity suggests that PRXamide peptides may serve as a dedicated conduit for transmitting signals between tissue layers. Conversely, within the endoderm, the principal signaling hub expresses the preprohormone G018128, which encodes GLWamide peptides (S5 Fig). This distinct expression pattern suggests that GLWamides may mediate communication in the opposite direction—from the endoderm back to the ectoderm.

Together, these observations support a model in which inter-tissue coupling relies on the selective deployment of discrete neuropeptide families, with each hub neuron preferentially producing and receiving specific peptides. This systematic organization likely enables precise and directional information flow across *Hydra´s* nerve nets.

## Attractor multistablity of *Hydra´s* neuropeptide network

Given the dense connectivity and prevalence of recurrent interactions, we also explored whether the inferred network architecture is capable of supporting specific forms of dynamical behavior. As demonstrated by Hopfield, synaptic circuits with distributed and recurrent excitatory connectivity can generate multistable attractor dynamics [43]. Indeed, attractor-like dynamics have been widely demonstrated in the nervous systems of many species, including mammalian cortex [44]. Attractor neural networks endow neural circuits with the capabilities to implement memory stated [45,46], solve optimization problems, computing head-direction in moving animals, implement pattern completion [47], and, more generally, serve as a Turing-complete computational "grammar" [48]. While these models have always relied on synaptic circuits, here, using a simplified model of the neuropeptide-based chemical connectome

in *Hydra*'s nerve net, we demonstrate strong evidence of dynamical convergence to two stable attractor states, both for an individual hub neuron and for the entire network (Fig 7). This multistability demonstrates that the architecture inferred from a purely chemical peptide–receptor interactions is not only structurally dense and recurrent but also dynamically capable of supporting multiple stable signaling configurations, a property consistent with flexible and state-dependent communication modes in the *Hydra* nervous system that could be used to implement specific behaviors and fixed action patterns.

## Limitations of our study and future work

Although based on systematic genomic and transcriptomic data [14,22], our study is purely computational and thus relies entirely not only on the quality of the datasets but also on the accuracy of the algorithms and AI tools for the identification of binding sites. First, our identification of GPCRs is done solely on the basis of homology – this is useful in a first attempt to find candidate GPCRs but can lead to novel or more species specific and unique GPCRs to be left out and not determined. A further structural analysis of all transcripts could be made in the future in which instead of using homology this structural analysis can be done allowing more depth and the discovery of novel or more divergent GPCRs. Moreover, while we use a set of computational tools that is widely accepted, we caution the reader that our results need biological validation. Three different levels of validation appear necessary: a biochemical one, confirming the predicted binding partners, an immunochemical one, confirming the expression of neuropeptides and proteins in specific cell types and thus allowing to match the transcriptome to the expressed proteome, and a functional one, demonstrating that the predicted neuropeptide-GPCR pairs binding actually leads to a significant communication between the emitting and the receiving neurons. Further, it would be of interest to include a epigenetic mapping data [22] and to locate the genes in terms of coregulation possibilities to understand how transcription factors can influence and play a role in shifts of the network and if this is through specific location orientation of processors and combinations. Ideally, to provide a context and comparison with synaptic network, this validation could be accompanied by connectomics approaches, either with ultrastructural or light level [49]. Thus, our results should be viewed solely as an initial guideline, or roadmap, of *Hydra*'s neuropeptide chemical network and could lead to a host of follow up studies that, given the complexity of the network that we uncover and the large effort involved in the biological validation and connectomics, fall outside the scope of this study.

## Evolutionary implications

The distributed neuropeptide network provides behavioral and evolutionary insights. The possibility to code complex behavior with a single neuropeptide has been demonstrated in species ranging from the effect of Hym 248 in *Hydra*'s somersaulting [17] to the control of mating and maternal behavior by oxytocin in humans [50,51]. Our clustering results provide insight into the evolutionary relationships between *Hydra* GPCRs and neuropeptide receptors from other organisms [11,39,52], reinforcing the hypothesis that core neuropeptide-GPCR interactions are conserved across cnidarians. Moreover, we find the relatively high numbers of neuropeptides and GPCRs in *Hydra* surprising, as it rivals those of centralized nervous system-bearing species. The complex chemical network that we have uncovered also provides a computationally very rich space to build dynamical systems, consistent with the hypothesis that the original nervous systems were based on neuropeptide signaling [10,28] which later, in bilaterians, became specialized for larger body sizes and speed by the evolution of chemical synapses [53]. Finally, the fact that the chemical network that we uncover is very dense, distributed and recurrent, enabling communication across all neurons, suggests that Hopfield-style attractor neural networks with widespread connectivity [45] can be built, not only with traditional synaptic wiring diagrams, but with equally connected chemical networks. The potential existence of stable or semi-stable attractor states could endow the nervous system of *Hydra* with the ability to encode a specific behavior in response to varying environmental conditions.

## Methods and models

### Datasets used

To identify the neuropeptide and GPCR complement of *Hydra vulgaris*, we used genome data to detect full-length receptor and precursor protein sequences, including transmembrane topology essential for GPCR classification, while transcriptome data were used to determine cell-type–specific expression and to validate that genome-derived candidates were transcriptionally supported. GPCRs and neuropeptide precursors were first predicted from the genome, and only those with evidence of neuronal expression in the single-cell transcriptome were retained. Structural receptor–peptide binding predictions were performed using genome-derived protein sequences, and the resulting interaction scores were integrated with transcriptome-based expression levels to construct the cell-type–resolved signaling network and subsequent dynamical models. The transcriptome data can be found under https://doi.org/10.5061/dryad.v5r6077 ˙(Siebert et al, [54] and the genome used can be found BioProject ID PRJNA816482 – deposited by the Juliano Lab.

### GPCR

To identify potential neuropeptide receptors from the GPCR family in the *Hydra vulgaris* genome we obtained the full sequence alignment of the class A GPCRs (PF00001) was obtained from the PFAM database and then made a Hidden Markov Model (HMM) with hmmer-3.1b2 [Eddy, 2011], which was then used to mine the *Hydra vulgaris*. Transcriptome to determine potential GPCRs. All GPCR protein sequences are provided in S1 File. The obtained sequences were analyzed using Phobius [Käll et al., 2007] to predict the number of transmembrane domains and only sequences with a minimum of four and maximum of nine transmembrane (TM) domains were kept for further analyses. We then used previous GPCR sequences determined for a phylogeny analysis of GPCRs across different clades (see ([20],Cnidaria: *N. vectensis, Alatina alata, C. cruxmelitensis, C. hemisphaerica, C. rubrum, E. pallida, H. vulgaris, Polypodium hydriforme, R. esculentum*. Bilateria: *D. melanogaster, C. elegans, Homo sapiens, P. marinus, P. dumerilii, Saccoglossus kowalevskii*. Placozoa: *Hoilunga hongkongensis, Trichoplax adhaerens*. Porifera: *Amphimedon queenslandica, Ephydatia muelleri, Oscarella carmella, Sycon ciliatum, Tethya wilhelma*, Ctenophora: *Pleurobrachia bachei, Mnemiopsis leydi, Hormiphora californiensis*) and the filasterian *Tunicaraptor unikontum*.[20] to look at the relationship between the obtained proteins and other characterized GPCRs across species by utilizing the CLANS software [55–57].The initial all-vs-all BLAST file was created using the online clans toolkit (https://toolkit.tuebingen.mpg.de/tools/clans), with the default BLOSUM62 scoring matrix and BLAST HSP's extracted up to E-values of 1e-4. The sequences were then clustered using the CLANS desktop version with a P-value cutoff of 6e-38. This threshold preserved clear and biologically meaningful clustering relationships between the *Hydras* GPCRs and the experimentally validated *Nematostella vectensis* neuropeptide receptors. Because *Nematostella* remains the only cnidarian with functionally confirmed receptor–ligand pairs, using a cutoff that maintains these inter-species relationships provided a justified and biologically grounded criterion. Sequences were then visualized and color-coded according to taxonomy and, for *Nematostella*, according to deorphanization status.

### Neuropeptide precursor search

We defined a putative secretome by using the computational tools SignalP 4.0 webtool [58] to determine all sequences in the transcriptome that had a putative signal sequence on them. This secretome (computationally defined) was combined with the results of pattern searches as described before in Thiel et al., 2021 [20], based on repetitive cleavage sites to search for novel precursors. The resulting sequences were then manually checked and aligned for occurrence of similar motifs between these cleavage sites.

## Matching peptides and GPCRs

We adapted the AF2–DeepTMHMM prediction framework described by Teufel et al. (2023), a peer-reviewed method originally validated on human neuropeptide–GPCR interactions. In their benchmarking, the true receptor for 11 independent peptides was consistently ranked within the **top 25 predictions out of >1,000 candidate GPCRs**, demonstrating robust performance in large receptor libraries. In our system, the search space is substantially smaller, and selecting the **top 10%** of predictions represents an even stricter threshold than the effective ranking window used in the Teufel et al. validation. Because experimentally confirmed false interactions are not available for *Hydra*, classical accuracy metrics cannot be computed; however, employing a validated prediction system alongside a conservative top-ranking cutoff provides a justified and stringent strategy for identifying high-confidence receptor–peptide pairs. Code available on GitHub https://github.com/fteufel/alphafold-peptide-receptors.

## Connectivity matrix

Matrices were constructed to represent connectivity between cells by calculating the expression of matching receptor and peptide pairs across all cells in the scRNAseq *Hydra vulgaris* atlas published by [21]. For each pair of cells, the expression levels of the corresponding receptor (in the potential receiver cell) and the peptide (in the potential sender cell) were multiplied to obtain a connectivity score which was combined with the AF score between them. If a pair was not scored as a binding pair this results in 0. These pairwise scores were compiled into a cell-to-cell connectivity matrix for each peptide-receptor pair.

No physical constraints (e.g., distance or proximity) were incorporated into the connectivity matrices, as *Hydra vulgaris* is an extremely mobile and contractile organism, and thus it must be assumed that physical positioning either does not play a major role or that its influence is highly dynamic. Modeling such physical constraints would require a more complex, dynamic model that captures tissue flexibility and continuous contraction behavior.

Each matrix was then analyzed for shared features, including the presence of hub cells with high connectivity, clusters of strongly connected cells, and the distribution of connection strengths. All individual matrices—one per peptide-receptor pair—were then integrated into a comprehensive connectivity network by summing scores across all peptide-receptor combinations. Finally, to facilitate biological interpretation, the connectivity matrix was collapsed to the level of cell types by aggregating cells based on their annotation. For each cell type pair, the sum of all connections (i.e., summed connectivity scores) was calculated, representing the potential for peptide-mediated signaling between those types. Rich Club analysis was done by sub setting a random 2% across all neurons to simulate a more realistic *Hydra vulgaris* size around 250 neurons and analyzing the connections in this network and the sub-structure formation.

## Multistability analysis

To assess whether the receptor–peptide signaling network supports multiple stable activity states, we constructed a recurrent dynamical system using the weighted adjacency matrix derived from our receptor–ligand inference pipeline (see Code and Data Availability). The final directed connectivity matrix represents the summed signaling influence between all neuronal and non-neuronal cell types.

Before simulation, the connectivity matrix was normalized by scaling all synaptic weights such that the magnitude of the dominant eigenvalue matched a fixed gain factor (1.3). This ensured that the system operated in a nonlinear regime capable of producing rich dynamics without saturating.

Network dynamics were simulated using a standard continuous-time rate-based model of the form:

$$\tau \frac{dx}{dt} = -x + \tanh(Wx),$$

where $x$ is the vector of cell-type activities, $W$ is the connectivity matrix, and $\tanh(\cdot)$ is a saturating nonlinearity. The system was integrated numerically for 3,000 time steps using small fixed increments.

To evaluate multistability, the network was initialized from 40 independent random activity states, each sampled from a uniform distribution over $[-1, 1]^N$. For each initialization, the full activity trajectory was recorded and the final state was extracted after convergence.

Distinct attractor basins were identified by comparing the final activity vectors across initial conditions. To visualize these end-states and assess the number of attractors, we embedded the final activity vectors into a low-dimensional space using principal component analysis (PCA). Clustering of final states in PCA space indicates the presence of multiple stable fixed points in the signaling network.

Highlights:

- Neuronal cell type-specific neuropeptide and GPCR expression in *Hydra vulgaris*

- Dense and recurrent neuropeptide-GPCR chemical interaction network with hubs

- Extensive ectoderm-endoderm neuropeptide signaling

- Distributed and recurrent chemical network can implement stable circuit attractors

## Supporting information

**S1 Fig. (Left) Weighted degree distribution showing the total strength of connections (considering connection weights) for all nodes, separated into in-degree (blue) and out-degree (orange).** The weighted out-degree distribution is more dispersed and exhibits higher variance, indicating that some neurons receive disproportionately strong inputs compared to others. The weighted in-degree distribution shows a bimodal pattern, suggesting the presence of specialized hub-like neurons with particularly high incoming signals. (Right) Unweighted degree distribution showing the count of connections regardless of strength (i.e., the number of edges), also separated into in-degree (blue) and out-degree (orange). The incoming distribution is more compact indicating a more uniform spread of connectivity when only the presence or absence of connections is considered.
(TIFF)

**S2 Fig. Bar plots display the average weighted degree (i.e., the sum of connection weights) for each cell type.** The blue bars represent neuron (en) subtypes, while the orange bars represent epithelial (ec) subtypes. This analysis highlights the differences in connectivity strength between cell types, revealing that many ec subtypes exhibit higher average weighted degrees compared to en subtypes.
(TIFF)

**S3 Fig. Rich Club Analysis.** High degree of interconnectedness and communication between individual neurons even at high thresholds – observe stability in the beginning which slowly drops and then peaks again revealing a subset of highly interconnected neurons.
(TIFF)

**S4 Fig. Distribution of neuron types in the top 10% richest nodes of the *Hydra vulgaris* neural network, as defined by the strongest connections.** The y-axis indicates the number of neurons of each type, while the x-axis labels denote specific neuronal subtypes (e.g., ec3C, ec5, ec3B, ec4). This analysis reveals a strong dominance of ectodermal neuron subtypes (ec3C, ec5, ec3B, ec4) in the rich club, highlighting their role as key communication hubs. Endodermal neurons (en1 and en2) and some less prominent subtypes (ec3A) contribute minimally to the rich club, suggesting a more

peripheral role in network connectivity. Overall, this composition underscores the importance of specific ectodermal neurons in forming the highly interconnected core of *Hydra vulgaris* neuropeptide network.
(TIFF)

**S5 Fig. Peptide motif diversity across Hydra preprohormone genes.** Bar plot showing the number and types of predicted peptide motifs encoded by some of the preprohormone genes. Colors indicate distinct peptide motif families, including GLWamide, GRFamide, GRYamide, GYGamide, INC, KVamide, LTRamide, LPW, PMIE, PRXamide, and peptides assigned to the "Other" category. Each bar represents one preprohormone gene, and stacked segments indicate the number of peptides belonging to each motif class. The plot highlights strong family-specific biases—for example, GLWamide-rich preprohormones (HVAEP10-G018128) and GRFamide-dominated preprohormones (HVAEP9-G017227, HVAEP9-G017228)—illustrating the functional specialization of different peptide-encoding loci.
(TIFF)

**S1 File. Contains all the sequences of the GPCR receptors determined in a fasta format that were used for this study.** These can be used as input for the code if needed to predict binding between GPCRs and peptides.
(FILE)

**S2 File. This file contains the amino acid sequences of all neuropeptide precursor candidates identified in this study, with the predicted peptide regions indicated within each precursor.** It includes the same information shown in Fig 1, provided here in an extractable format for reuse and further analysis.
(DOCX)

**S1 Table. This table contains the names, transcript identifiers, and amino acid sequences of all predicted final neuropeptides identified in this study, along with the predicted communication pathways.** The sequences of the final peptide products can be used as input for computational binding predictions.
(XLSX)

## Acknowledgments

We thank past and present members of the Yuste lab for help and fruitful discussions and helpful comments.

## Author contributions

**Conceptualization:** Johanna de la Cruz Rothenfusser, Rafael Yuste.

**Data curation:** Johanna de la Cruz Rothenfusser.

**Formal analysis:** Johanna de la Cruz Rothenfusser, Luis Alfonso Yáñez-GuerraLuis.

**Funding acquisition:** Rafael Yuste.

**Investigation:** Johanna de la Cruz Rothenfusser, Rafael Yuste.

**Methodology:** Luis Alfonso Yáñez-GuerraLuis, Felix Teufel.

**Project administration:** Rafael Yuste.

**Resources:** Johanna de la Cruz Rothenfusser, Felix Teufel, Rafael Yuste.

**Software:** Luis Alfonso Yáñez-GuerraLuis, Felix Teufel.

**Supervision:** Luis Alfonso Yáñez-GuerraLuis, Rafael Yuste.

**Validation:** Johanna de la Cruz Rothenfusser.

**Visualization:** Johanna de la Cruz Rothenfusser.

**Writing – original draft:** Johanna de la Cruz Rothenfusser, Rafael Yuste.

**Writing – review & editing:** Johanna de la Cruz Rothenfusser, Luis Alfonso Yáñez-GuerraLuis, Rafael Yuste.

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
