## [Decision Letter · Decision Letter 0]

26 Aug 2025

PCOMPBIOL-D-25-01397

Dense and distributed neuropeptide network in Hydra vulgaris

PLOS Computational Biology

Dear Dr. de la Cruz,

Thank you for submitting your manuscript to PLOS Computational Biology. After careful consideration, we feel that it has merit but does not fully meet PLOS Computational Biology's publication criteria as it currently stands. Therefore, we invite you to submit a revised version of the manuscript that addresses the points raised during the review process.

Please submit your revised manuscript within 60 days Oct 26 2025 11:59PM. If you will need more time than this to complete your revisions, please reply to this message or contact the journal office at ploscompbiol@plos.org. Please include the following items when submitting your revised manuscript:

We look forward to receiving your revised manuscript.

Kind regards,

Eduardo Jardón-Valadez

Academic Editor

PLOS Computational Biology

Joseph Ayers

Section Editor

PLOS Computational Biology

**Additional Editor Comments:**

Dear Dr. Johana de la Cruz,

We sincerely appreciate your interest in publishing your research in PLOS Computational Biology. After careful evaluation, we believe that your manuscript would benefit from a more thorough revision in light of the suggestions and concerns raised by the invited reviewers.

In particular, we kindly ask you to consider addressing the potential role of epigenetic factors in gene expression, as well as how the proposed network topology might adapt to varying environmental conditions.

We look forward to receiving your revised manuscript at your earliest convenience.

With best regards,

Eduardo Jardón

**Journal Requirements:**

At this stage, the following Authors/Authors require contributions: Johanna de la Cruz, Luis Alfonso Yañez, Felix Teufel, and Rafael Yuste. Please ensure that the full contributions of each author are acknowledged in the "Add/Edit/Remove Authors" section of our submission form.

4) We notice that your supplementary Figures are included in the manuscript file. Please remove them and upload them with the file type 'Supporting Information'. Please ensure that each Supporting Information file has a legend listed in the manuscript after the references list.

Potential Copyright Issues:

i) Figure 5 appears to have been adapted from a previously published figure. Please provide written permission from the copyright holder to publish this under our CC-BY 4.0 license, or remove the figure / replace the image. Please note we do not recommend using standard request forms available on Publishers' websites, as they grant single use rather than republication under an open access license.

6) We note that your Data Availability Statement is currently as follows: "all data either uploaded or given in the linked github". Please confirm at this time whether or not your submission contains all raw data required to replicate the results of your study. Authors must share the “minimal data set” for their submission. PLOS defines the minimal data set to consist of the data required to replicate all study findings reported in the article, as well as related metadata and methods (https://journals.plos.org/plosone/s/data-availability#loc-minimal-data-set-definition).

7) Please amend your detailed Financial Disclosure statement. This is published with the article. It must therefore be completed in full sentences and contain the exact wording you wish to be published.

8) Your current Financial Disclosure states, "Yes ↳ Please add funding details. NSF (2203119) and the Vannevar Bush Faculty Award (ONR N000142012828). ↳ Please select the country of your main research funder (please select carefully as in some cases this is used in fee calculation). UNITED STATES - US".

However, your funding information on the submission form indicates no funds received .

Please indicate by return email the full and correct funding information for your study and confirm the order in which funding contributions should appear. Please be sure to indicate whether the funders played any role in the study design, data collection and analysis, decision to publish, or preparation of the manuscript.

9) Please send a completed 'Competing Interests' statement, including any COIs declared by your co-authors. If you have no competing interests to declare, please state "The authors have declared that no competing interests exist". Otherwise please declare all competing interests beginning with the statement "I have read the journal's policy and the authors of this manuscript have the following competing interests"

**Reviewers' comments:**

Reviewer's Responses to Questions

**Comments to the Authors:**

Reviewer #1: This study constructs a chemical connectome for Hydra vulgaris, offering a well-curated, organism-wide hypothesis space for neuropeptide signaling in a basal metazoan. The resulting resource is both timely and of practical value to the field. Here are my comments:

1. While the network is dense and recurrent, no dynamical modeling is presented. The authors are encouraged to incorporate a minimal dynamical model to demonstrate the potential for multistability within the inferred signaling architecture.

2. There are inconsistencies in the reported numbers of mature peptides and GPCRs. The Abstract and Results sections alternately report 61 and 65 mature peptides (with Fig. 1 caption stating “61”), while the number of GPCRs varies between 65 (Results) and 63 (Methods). These discrepancies should be clarified and corrected across the manuscript.

3. Please standardize terminology and address editorial issues throughout the manuscript. This includes:

Consistent use of the species name (Hydra vulgaris),

Standardized neuropeptide family nomenclature (e.g., FR, GRF, RFamide),

Correction of typographical inconsistencies (e.g., “GYGY” vs. “GGYG amide”),

Proper use of English possessives (“Hydra’s” instead of “Hydra´s”),

Standardization of month names in references (e.g., “December” instead of “Desember”; “June” instead of “Junie”).

Reviewer #2: De La Cruz et al. present a computational study of neuropeptide networks in Hydra Vulgaris. The study is interesting and the findings made are potentially very important for comparative neuroscience, but I believe the analyses need to be better documented and justified to be suitable for PLOS Computational Biology. Ideally, in my view, the findings could also be interpreted in more detail with specific cases (e.g. of a putative key network hub) highlighted.

# Major points

- the bioinformatic analyses are not sufficiently documented e.g. what versions of tools were used? (sometimes stated, often not). I cannot see code for e.g. the HMM, SignalP, or clans analyses in the Jupyter notebook provided (I haven’t assessed it line by line but couldn’t find it). Some main headers and/or splitting the notebook into different analyses would be helpful, along with a more detailed ‘readme’ file.

- thresholds chosen need to be mentioned and justified (e.g. why 6e-38 for the CLANS clustering?)

- the nature of the underlying data was not clearly explained or linked to (the ID in the relevant database e.g. SRA should be given). For instance the “transcriptome” repeatedly referred to appears to be derived from single cell transcriptomic experiments but the nature of this data and how much of it was used is not clear. The role of the epigenetic map of the Hydra genome cited is also not clear.

- it is not clear that the search for homologs based on HMMs is indeed “comprehensive”, given the evolutionary distances involved in comparing cnidarians to other groups. A more sensitive search approach such as structure-based approach (ideally with a sensitive method) may well find more relevant GPCRs or neuropeptides. This could be noted as a point for further investigation. Can the implicit claim that all relevant GPCRs have been found been justified? (e.g. were any at the borderline of the thresholds used, which would imply that it is likely that some were missed)

- the meaning/interpretation of the network properties reported was not sufficiently explained, in my view - how does it relate to other similar networks? Are the results expected? The novelty of the results for Cnidarians is also not quite clear although it seems the claims may be very novel. If so, this can be stated more precisely.

- I think more can be said about the putative connection hubs discovered from the network, as this seems a key result. E.g. do neuropeptides/receptors involved in hubs differ from others?

## Minor comments

- the work of Cajal & Sherrington is not cited.

- Figure 2 - the meaning of the red box is not clear to me, and likewise the titles “GPCR families” and “putative neuropeptide binding GPCR families” are not clear for this figure - which subsets/objects do each refer to?

- Figures e.g. Figure 5 need to be at higher resolution.

- The writing is generally very good but there are a few grammar/spelling issues to fix - e.g. “effectivity of the algorithms”, “reciprocals signalling pathways”, “show low to none synaptic connectivity”.

- I don’t believe the binding predictions are likely to be accurate, but may be wrong. Is there experimental evidence for a system which is at all similar, using the Alphafold + DeepTMHMM method? Quantifying the expected accuracy here would be useful.

**Have the authors made all data and (if applicable) computational code underlying the findings in their manuscript fully available?**

Reviewer #1: None

Reviewer #2: **No:** The code provided is not clearly organised and does not appear to all be available

PLOS authors have the option to publish the peer review history of their article (what does this mean? ). If published, this will include your full peer review and any attached files.

**Do you want your identity to be public for this peer review?** For information about this choice, including consent withdrawal, please see our Privacy Policy .

Reviewer #1: No

Reviewer #2: No

**Figure resubmission:**
---

## [Decision Letter · Decision Letter 1]

17 Feb 2026

Dear Ms de la Cruz,

We are pleased to inform you that your manuscript 'Dense and distributed neuropeptide network in the nerve net of Hydra vulgaris' has been provisionally accepted for publication in PLOS Computational Biology.

Best regards,

Sarah Mayo

Staff Admin

PLOS Computational Biology

Joseph Ayers

Section Editor

PLOS Computational Biology

Reviewer's Responses to Questions

**Comments to the Authors:**

Reviewer #1: I have no other concerns.

Reviewer #2: Thank you for responding to all suggestions, and making some important updates to the work.

I have one final comment - in the discussion section, based on my recommendation, it has now been suggested that a structural analysis could be conducted in future in addition to a "homology" analysis - however, structural comparison, as with HMMs, is a method of homology search, it is just structure-based rather than sequence-based homology search. I suggest correcting this with minor re-wording.

**Have the authors made all data and (if applicable) computational code underlying the findings in their manuscript fully available?**

Reviewer #1: None

Reviewer #2: Yes

PLOS authors have the option to publish the peer review history of their article (what does this mean? ). If published, this will include your full peer review and any attached files.

**Do you want your identity to be public for this peer review?** For information about this choice, including consent withdrawal, please see our Privacy Policy .

Reviewer #1: No

Reviewer #2: **Yes:** Zachary Ardern

---

## [Editor Report · Acceptance letter]

PCOMPBIOL-D-25-01397R1

Dense and distributed neuropeptide network in the nerve net of Hydra vulgaris

Dear Dr de la Cruz Rothenfusser,

I am pleased to inform you that your manuscript has been formally accepted for publication in PLOS Computational Biology. Your manuscript is now with our production department and you will be notified of the publication date in due course.

With kind regards,

Judit Kozma
